# Usefulness of Practitioner-Led Pancreatic Cancer Screening

**DOI:** 10.3390/diagnostics12092257

**Published:** 2022-09-19

**Authors:** Kazuya Matsumoto, Yoshinori Kushiyama, Akio Izumi, Koji Ohnishi, Masahiko Miura, Yasufumi Ohuchi, Ikuko Hori, Tomonori Nakamura, Kotaro Hori, Kenji Koshino, Junko Kobayashi, Nagisa Yoshino, Ushio Hoshino, Takekiyo Okumura, Takashi Tanimura, Shinsuke Tanaka, Shino Tanaka, Tadashi Nabika, Tatsuaki Nozu, Yutaka Wakatsuki, Syunsuke Katayama, Shizue Yoshioka, Kenichi Ito, Akiko Uchida, Noriji Yuhara, Hisashi Noma, Hajime Isomoto

**Affiliations:** 1Irisawa Medical Clinic, Matsue 690-0025, Japan; 2Pancreatic Cancer Countermeasures Matsue Medical Association, Matsue 690-0048, Japan; 3Division of Medicine and Clinical Science, Department of Multidisciplinary Internal Medicine, Faculty of Medicine, Tottori University, Yonago 683-8504, Japan; 4Department of Data Science, The Institute of Statistical Mathematics, Tachikawa 190-8562, Japan

**Keywords:** pancreatic cancer, abdominal ultrasound, cancer screening

## Abstract

The 5-year survival rate for pancreatic cancer has improved (10%) but remains worse than that for other cancers. Early pancreatic cancer diagnosis is challenging, and delayed diagnosis can delay treatment, which impairs survival. Practitioners do not promptly refer cases to a general hospital, causing delayed discovery. Herein, we aimed to examine the usefulness of the Pancreatic Cancer Project in Matsue, whose objective is to detect pancreatic cancer in patients presenting at any medical institution in Matsue City. Clinical data were extracted from medical records, and abdominal ultrasonography and tumor marker blood level assessments were performed (*n* = 234; median age, 71 [range, 41–94] years; 51% male). Cases with abnormal abdominal ultrasonography or blood test findings were referred for specialist imaging and followed up. The pancreatic cancer detection rate was 6.0% (*n* = 14); all cases were referred to a general hospital by practitioners within 1 month. Patients had stage IA (*n* = 1), IIA (*n* = 6), IIB (*n* = 2), III (*n* = 1), and IV (*n* = 4) disease. Overall, pancreatic cancer could be detected at an earlier stage (I–II), but referral to a general hospital by visiting practitioners should be prompt. The Pancreatic Cancer Project in Matsue may help improve the detection and prognosis of pancreatic cancer.

## 1. Introduction

The 5-year survival rate for pancreatic cancer has improved (10%) but remains worse than that for other cancers [1]. Delayed diagnosis and treatment are the biggest contributors to poor survival; early detection of pancreatic cancer is challenging. Indeed, data from specialized facilities, such as university hospitals, indicate a 5-year survival rate of 2.0–4.7% in patients with stage I–II disease, accounting for 66.2% of cases (15,601/23,582) [2]. These values suggest that stage IV pancreatic cancer may account for ≥70–80% of cases, some of which present at facilities that lack pancreatic cancer specialists.

Factors that delay the diagnosis of pancreatic cancer include practitioner overconfidence in abdominal ultrasonography (US), failure to perform follow-up examinations with other modalities even if the pancreas is poorly visualized, and tardiness caused by practitioners at the referring hospitals. Only 25% of patients with stage 0–I disease present with symptoms [3]. However, 82% of patients with pancreatic tumors less than 2 cm may present with isolated symptoms [4], making it difficult to detect early stage (0–I) pancreatic cancer at the primary care level.

Although pancreatic cancer may be diagnosed earlier than at stage III–IV, which represents advanced disease, there is currently no established method for diagnosing it at stage 0–I. In addition, previous studies have reported early detection rates of pancreatic cancer in various regions of Japan [5,6]; however, follow-up studies on those cases are lacking.

The Matsue Medical Association launched the Pancreatic Cancer Project in Matsue in June 2018 (UMIN000035462). The objective of the project is to increase the rates of tertiary care referrals among patients suspected of pancreatic cancer, including referrals to centers that do not specialize in pancreatic cancer care. Herein, we aimed to verify the contribution of the Matsue City Pancreatic Cancer Project to the early detection of pancreatic cancer.

## 2. Materials and Methods

### 2.1. Study Design and Ethics Approval

Patients aged ≥ 40 years were eligible for this study. Risk factors included clinical symptoms (abdominal pain, back pain, weight loss, and abdominal bloating), type II diabetes mellitus, smoking, drinking > 60 g of ethanol per day, family history of pancreatic cancer, body mass index of ≥30 kg/m^2^, chronic pancreatitis, and intraductal papillary mucinous neoplasm (IPMN). Cases with two or more clinical symptoms or risk factors were selected for closer examination; thereafter, abdominal US evaluations and blood tests were performed. Blood samples were examined for tumor markers, such as carbohydrate antigen (CA19-9), Span-1, and DUPAN2. Cases with abnormal abdominal US or blood test findings were referred to a general hospital and examined by dynamic computed tomography (CT), magnetic resonance cholangiopancreatography(MRCP), or endoscopic US (EUS) scanning. Abdominal echo was performed by gastroenterologists as well as doctors of other specialties who had especially attended a 30 min pancreatic echo hands-on seminar. Patients with any abnormal findings and pancreatic cancer diagnoses were followed up, and their outcomes were evaluated. The study complies with ethical guidelines as mentioned in the Declaration of Helsinki.

### 2.2. Data Collection

The Matsue Red Cross Hospital Institutional Review Board (protocol code: 382; date of approval: 21 May 2018) approved this study involving 234 consecutive patients who visited a medical institution in Matsue City between June 2018 and June 2021 (median age, 71 [range, 41–94] years; 51% males). The sample included patients with pancreatic ductal cell adenocarcinoma (*n* = 14; 6%), pancreatic cysts, main pancreatic duct dilation, or elevated tumor markers (*n* = 152; 65%). A total of 60 patients presented without abnormalities, while eight presented with “other” abnormalities (Table 1).

### 2.3. Statistical Analyses

Logistic regression models were used to identify prognostic factors in pancreatic cancer. First, marginal associations among potential predictors were evaluated in univariate logistic regression analyses. Second, multivariate logistic regression models were created to assess individual and combined effects of potential predictor variables. Factors identified as statistically significant (*p* < 0.05) in the univariate analyses were included as explanatory variables in the multivariate analyses. For missing data, we used multiple imputations by chained equation with 200 imputations. Predictive scores obtained in the multivariate models were used to create the receiver operating characteristic (ROC) curve; the area under the ROC curve (AUC) was calculated as the discriminant measure. In addition, we computed optimism-corrected AUC values by using the Harrell bootstrap bias correction method. All analyses were performed using R version 4.2.0 (R v4.2.0 https://cran.r-project.org/bin/windows/base/old/ (accessed on 22 April 2022)). All *p*-values were two-sided.

## 3. Results

The pancreatic cancer detection rate was 6.0% (*n* = 14), and all cases were referred to general hospitals by practitioners within 1 month. The patients had stage IA (*n* = 1), IIA (*n* = 6), IIB (*n* = 2), III (*n* = 1), and IV (*n* = 4) disease. The modalities used to detect pancreatic cancer were US (*n* = 9), dynamic CT (*n* = 3), and MRCP (*n* = 2). Clinical symptoms included abdominal pain, palpable mass, weight loss, abdominal bloating, and no symptoms in five, two, two, two, and three cases, respectively. In addition, 71.4% (10/14) of pancreatic cancer cases had some associated risk factors, and 85.7% (12/14) of them had high tumor marker levels (Table 2).

Among the 174 enrolled patients, 138 (79.3%) were followed up. Of these, 13 patients had pancreatic cancer at registration, and 125 with other pancreatic disorders were followed up. Four patients with pancreatic cancer died. Among patients with other pancreatic disorders that were followed up, one (0.8%) had branch-duct (BD)-IPMN and developed pancreatic cancer after 32 months of follow-up (Figure 1).

We found that Span-1 (odds ratio [OR], 15.752; 95% CI, 1.703–145.721; *p*-value, 0.015) and pancreatic tumors (OR, 37.534; 95% CI, 3.564–395.254; *p*-value, 0.003) were risk factors for pancreatic cancer diagnosis. In contrast, pancreatic cysts (OR, 0.391; 95% CI, 0.016–9.443; *p*-value, 0.563) did not predict pancreatic cancer diagnosis (Table 3). 

The risk score was calculated as follows: −6.18 + 0.66 × 1 + 1.62 × 14 + 2.76 × 15 + 0.99 × 16 − 0.9 × 18 + 3.63 × 19 (×1: abdominal pain, ×14: CA19-9, ×15: Span-1, ×16: DUPAN2, ×18: pancreatic cyst, ×19: pancreatic tumor). For a risk score of >0 points, the AUC for pancreatic cancer diagnosis was 0.955 (95% CI, 0.934–0.976) (Figure 2).

A 68-year-old man consulted a practitioner with a complaint of weight loss. Abdominal US examination showed no abnormal findings. The patient had a history of diabetes mellitus and smoking, which are risk factors for pancreatic cancer; thus, CA19-9, Span-1, and DUPAN2 levels were evaluated, revealing elevated values. The attending practitioner referred the patient to a general hospital for further examination. Plain abdominal CT scan showed no notable findings (Figure 3a), although a dynamic CT scan showed a mass with a major axis of 10 mm, which had a poor contrast effect in the tail of the pancreas (Figure 3b). The patient was diagnosed with pancreatic cancer.

## 4. Discussion

Most pancreatic cancer cases diagnosed by the Matsue City Pancreatic Cancer Project were stage IIA cases, with a 5-year survival rate of 30.2% [2]. This project enabled patients to consult with practitioners who could refer them for hospital evaluations at an early stage. In this study, abdominal US evaluation performed by the practitioner was not a useful diagnostic tool. However, blood tumor marker levels and lesions identified by dynamic CT scans were diagnostically useful. Identifying patients with these characteristics may improve the prognosis of pancreatic cancer.

In Japan, evaluations for tumor markers of pancreatic cancer such as carcinoembryonic antigen (CEA), CA19-9, Span-1, and DUPAN2, are covered by insurance. These markers are included in the Matsue City Pancreatic Cancer Project based on evidence from previous studies. Our previous data on the diagnostic ability assessment of tumor markers in 246 patients with pancreatic cancer showed that CA19-9, Span-1, DUPAN2, and CEA levels were associated with diagnostic accuracy values of 74.3%, 69.6%, 57.3%, and 45.5%, respectively (data not published). The number of tumor marker tests covered by insurance in Matsue is limited to three tests per month. Herein, Span-1 level was confirmed as a useful index in the risk score calculation. Pancreatic enzyme level evaluation may help detect early stage pancreatic cancer [7]; a combination of these markers may increase diagnostic accuracy.

In this study, pancreatic cysts were a negative predictor of pancreatic cancer. Although pancreatic cysts are frequently observed, they progress to pancreatic cancer in only a few cases. The frequency of pancreatic cancer with BD-IPMN is 2–10% [8]. In addition, the cumulative 5-year incidence rate of pancreatic cancer in patients with BD-IPMN who are not indicated for resection and are being followed up is approximately 6.9–8.8% [8,9]. In this study, the onset of pancreatic cancer was observed in IPMN cases after 32 months of follow-up. In 2 of 14 cases of pancreatic cancer, only pancreatic cysts were observed at the time of abdominal US. Given these findings, patients with pancreatic cysts should be followed up and monitored for the signs of cancer development, even if those signs are absent at the time of cyst detection.

Conventionally, the relationship between diabetes and pancreatic cancer was thought to be caused by pancreatic tissue destruction caused by concomitant pancreatitis caused directly by pancreatic cancer or by obstruction of the pancreatic duct, resulting in decreased insulin secretion. Recently, in addition to the mechanism described above, exosomes produced by pancreatic cancer are thought to exacerbate diabetes by inducing apoptosis and insulin resistance in pancreatic β cells [10]. As a basis for this, it has been reported that in patients undergoing pancreatic cancer surgery, although exacerbation of diabetes is expected, diabetes may actually improve [11,12]. Overall, 0.85% of new-onset secondary diabetes mellitus cases are associated with pancreatic cancer [13]. The risk of developing pancreatic cancer is significantly increased in both type 1 and type 2 diabetes mellitus [14,15], and pancreatic cancer is often detected at the time of diabetes mellitus onset or acute exacerbation [15]. Diabetes mellitus was not a significant risk factor for the development of pancreatic cancer in this study. However, this may be because at registration, it was not confirmed whether the cases were known cases of diabetes mellitus or represented those with acute exacerbation. Future studies should investigate the relationship of diabetes mellitus onset and exacerbation with pancreatic cancer risk, including the impact of blood glucose and glycated hemoglobin levels at the time of pancreatic cancer onset.

In this study, 4.1% of registered cases were unqualified cases. Many of the unqualified cases were due to misinterpretation of abdominal ultrasonography findings. If a patient is diagnosed with pancreatic cancer and is referred to a general hospital even though he/she is not eligible for registration, it will cause mental distress for the patient before the test result is obtained. In future research, it is important to inform the subjects of detailed examination through hands-on seminars and consider how to reduce unqualified cases. In this study, 28.9% of the follow-up cases were unfollowed. Since pancreatic cancer has also developed from the cases targeted for follow-up, it is also meaningful to hold regular public lectures related to pancreatic cancer screening and lectures for practitioners in order to maintain motivation of patients and practitioners.

In this study, no case of pancreatic cancer was confirmed with EUS. This finding may be associated with the availability of this modality in Matsue city. Some studies have suggested that EUS evaluation may help detect pancreatic cancer [16,17]; increased uptake of this modality may increase the rate of early-stage pancreatic cancer diagnosis.

Pancreatic cancer has a poor prognosis, but the prognosis tends to improve due to improvements in sample collection techniques, adjuvant therapy for resectable cases, and chemotherapy for unresectable cases. Regarding specimen collection, there are reports of improved accuracy of pancreatic juice cytology (PJC) [18,19], improved Endoscopic- ultrasound guided fine-needle biopsy (EUS-FNB) technique [20,21], and improved diagnostic performance by combining PJC and EUS-FNB [22]. In adjuvant chemotherapy for resected cases, in the PRODIGE 24/CCTG PA.6 trial, The median overall survival (OS) was 54.4 months in the modified-FOLFIRINOX group and 35.0 months in the gemcitabine group [23]. In a report of preoperative therapy for borderline cases, the PREOPANC trial compared upfront surgery with preoperative chemoradiation therapy with gemcitabine and immediate surgery. In patients who underwent resection, the median OS was 19.8 months in the immediate surgery group and 35.2 months in the preoperative chemoradiation group [24]. In chemotherapy for unresectable patients, the ACCORD11 trial was a Phase 3 trial comparing FOLFIRINOX and gemcitabine in patients with metastatic pancreatic cancer, with the primary endpoint of a median OS of 11.1 months in the FOLFIRINOX group, the gemcitabine group was 6.8 months [25]. Although the prognosis is improved regardless of the stage of the pancreatic cancer and the start of treatment, it is important to detect pancreatic cancer at an earlier stage in order to further improve the prognosis. Despite the high oversight rate of pancreatic cancer, although our study was conducted by primary-care-level physicians with no pancreatic cancer specialists, all cases were referred to the hospital within 1 month after visiting a general practitioner. In the follow-up cases, no unresectable pancreatic cancer due to oversight was observed. From these points, in addition to diagnosis, specimen collection, surgery, and chemotherapy, which are the cornerstones of pancreatic cancer treatment, we believe that practicing the method of The Pancreatic Cancer Project in Matsue at the primary care level will be a key to improving the prognosis of pancreatic cancer.

## 5. Conclusions

The Pancreatic Cancer Project in Matsue is useful for improving the prognosis of patients with pancreatic cancer. Early diagnosis of pancreatic cancer may be supported by pancreatic enzyme blood level assessments and uptake of EUS.

## Figures and Tables

**Figure 1 diagnostics-12-02257-f001:**
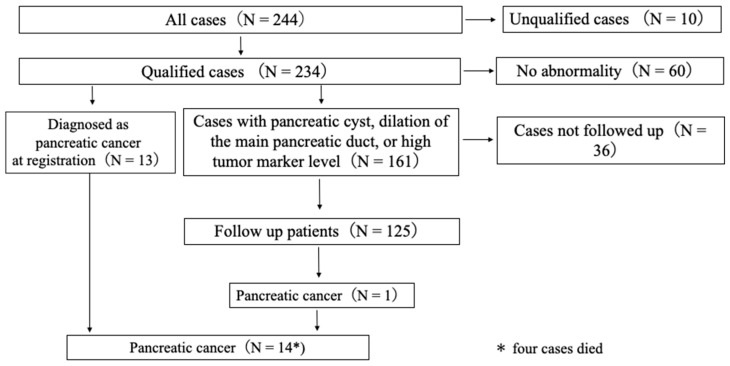
Patient participation flow and final diagnoses.

**Figure 2 diagnostics-12-02257-f002:**
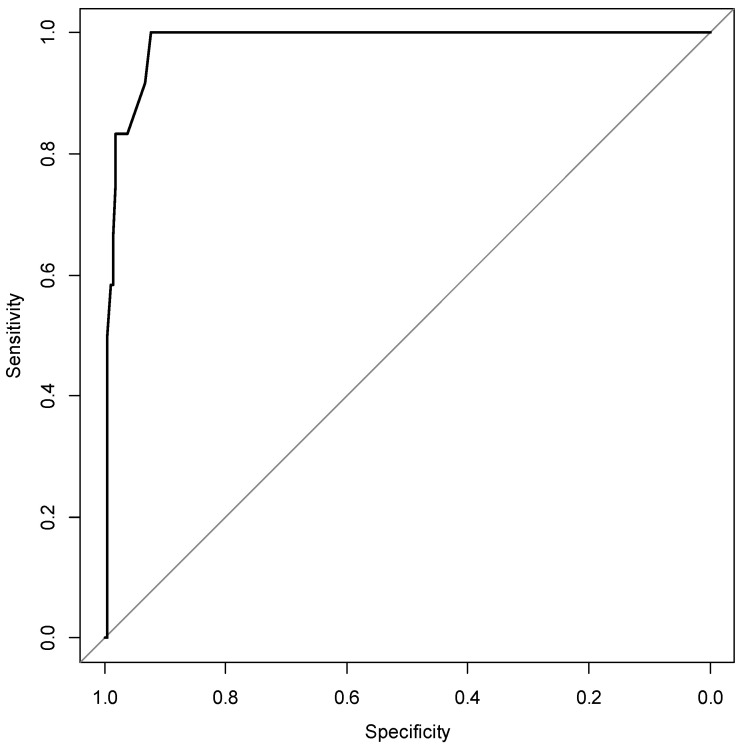
Predictive performance of the risk score for the diagnosis of pancreatic cancer. AUC, area under the curve; CI, confidence interval. AUC: 0.955 (95% CI: 0.934, 0.976).

**Figure 3 diagnostics-12-02257-f003:**
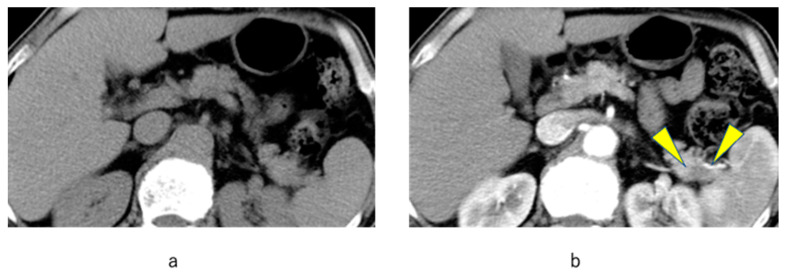
Imaging results indicating a case of pancreatic cancer successfully diagnosed based on the methods outlined in the Matsue City Pancreatic Cancer Project. (**a**) plain abdominal CT scan; (**b**) dynamic CT scan.

**Table 1 diagnostics-12-02257-t001:** Patient characteristics.

	All Cases (*n* = 234)
Age, in years	
Median age	71
Range	41–94
Sex	
Male	120
Female	114
Final diagnosis	
Pancreatic ductal adenocarcinoma	14
Pancreatic cyst or main pancreatic duct dilatation or high tumor marker levels	152
No abnormality	60
Others	8

**Table 2 diagnostics-12-02257-t002:** Characteristics, risk factors, imaging findings, and disease stages of patients with pancreatic cancer (*n* = 14).

Age (in years), Sex	Discovery Opportunity Modality	Symptoms	Risk Factors	High Level of Tumor Maker	Abdominal US Finding			Stage (UICC)
					MPD Dilatation	Pancreatic Cyst	Pancreatic Mass	
68 M	dyn CT	Body weight loss	Diabetes Smoking	All	−	−	−	IIA
85 F	MRCP	Palpable mass	None	CA19-9 Span-1	−	−	−	IIA
71 F	US	Abdominal bloating	None	CA19-9 DUPAN2	−	−	+	IIA
77 F	US	Abdominal bloating	IPMN	CA19-9 Span-1	+	+	+	IIB
60 F	US	Palpable mass	Smoking	CA19-9	−	−	+	III
87 M	US	None	Diabetes	CA19-9 Span-1	+	−	+	IA
64 F	dyn CT	Body weight loss	Smoking Drinking	CA19-9 Span-1	−	+	−	IIA
76 M	dyn CT	Abdominal pain	None	CA19-9 Span-1	+	−	−	IIA
85 F	US	Abdominal pain	Diabetes	Span-1 DUPAN2	−	−	+	IV
69 M	US	None	Smoking	Span-1 DUPAN2	+	−	+	IV
68 F	US	Abdominal pain	None	None	−	−	+	IIB
76 F	US	Abdominal pain	None	Span-1 DUPAN2	+	−	+	IV
66 M	US	Abdominal pain	Smoking Drinking	All	+	−	+	IV
63 M	MRCP	None	IPMN	None	−	+	−	IIA

MPD, main pancreatic duct; US, ultrasonography; UICC, Union for International Cancer Control; dyn CT, dynamic computed tomography; MRCP, magnetic resonance cholangiopancreatography; IPMN, intraductal papillary mucinous neoplasm.

**Table 3 diagnostics-12-02257-t003:** Predictors of pancreatic cancer.

	Odds Ratio	95% Confidence Interval		*p*-Value
×1 Abdominal pain	1.942	0.259	14.557	0.518
×14 CA19-9	5.042	0.57	44.59	0.146
×15 Span-1	15.752	1.703	145.721	0.015
×16 DUPAN-II	2.686	0.309	23.338	0.37
×18 Pancreatic cyst	0.391	0.016	9.443	0.563
×19 Pancreatic mass	37.534	3.564	395.254	0.003

## Data Availability

The data presented in this study are available on request from the corresponding author. The data are not publicly available due to privacy laws.

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
