# Peer review of "Usefulness of Practitioner-Led Pancreatic Cancer Screening"

_diagnostics, 2022, doi:10.3390/diagnostics12092257_

Round 1

Reviewer 1 Report

Paper theme is very interesting and should be published. 

Author Response

Thank you for your review. Since your comment was ``Paper theme is very interesting and should be published'', I don't have any particular comments.

Reviewer 2 Report

Since, chemical nature of pancreatic cancer markers used in the study is not clear except for CA19-9, the conclusion that "pancreatic enzymes" (Early diagnosis............uptake of EUS) should be monitored will be not be right. It may be suitably modified mentioning markers used in the study.

Author Response

Thank you for your suggestion. I added a reference to line 162 as you pointed out.  please confirm.

Reviewer 3 Report

The article is interesting and I found it informative as a physician who looks after pancreatic problems.

 I just have a couple of minor suggestions.  

1)     Line 156: “An our previous data on the diagnostic ability assessment of tumor 156 markers in 246 patients with pancreatic cancer showed ….”

Suggestion:  Do you want to start this statement using " Our...." or “In our previous data …”

2)     Correct grammar in lines 192, and 197.

Suggestion: In these 2 lines, do you want to use ‘unqualified cases’? I shall keep ‘U’ small for correct grammar.

Author Response

Thank you for your suggestion. As you pointed out, Line 156 states that I start this statement using " Our...." . I have corrected grammar in lines 192, and 197.

Please check them.
